# Complement activation assessed by C3bc and C5b-9 terminal complex as diagnostic biomarkers for deep vein thrombosis

Vårin Eiriksdatter Wikan[1]*, Øyvind Øverli[2], Thor Ueland[3,4,5], Tom Eirik Mollnes[6,7], Sigrid Kufaas Brækkan[1,4], Annika Elisabet Michelsen[3,5], Gholamreza Jafari Yeganeh[8], John-Bjarne Hansen[1,4], Ellen Elisabeth Brodin[2]

1 Department of Clinical Medicine, Thrombosis Research Group (TREC), UiT – The Arctic University of Norway, Tromsø, Norway, 2 Hematological Research Group, Division of Medicine, Akershus University Hospital, Lørenskog, Norway, 3 Research Institute of Internal Medicine, Oslo University Hospital (Rikshospitalet), Oslo, Norway, 4 Thrombosis Research Center (TREC), Division of Internal Medicine, University Hospital of North Norway, Tromsø, Norway, 5 Institute of Clinical Medicine, University of Oslo, Oslo, Norway, 6 Research Laboratory, Nordland Hospital, Bodø, Norway, 7 Department of Immunology, Oslo University Hospital and University of Oslo, Oslo, Norway, 8 Department of Emergency Medicine, Akershus University Hospital, Lørenskog, Norway

* varin.e.wikan@uit.no

## Abstract

### Background

The overuse of compression ultrasound procedures on patients with suspected deep vein thrombosis compromise the cost-effectiveness of deep vein thrombosis management and emphasize room for improvement of the current diagnostic algorithm. As the complement system and hemostasis are comprehensively intertwined, we aimed to investigate complement activation products as diagnostic tests for acute deep vein thrombosis alone or together with D-dimer.

### Methods

We performed a cross-sectional study using consecutive sampling of outpatients referred to the emergency department with suspected first-time deep vein thrombosis of the lower limbs, to investigate the diagnostic utility of the index tests C3bc and C5b-C9 terminal complex (TCC) with compression ultrasound as reference standard. For comparison of receiver operating characteristics, analyses were also performed for D-dimer and C-reactive protein in addition to analyses for the index tests on a D-dimer positive patient subgroup.

### Results

Of the 366 included patients, 103 had deep vein thrombosis. The calculated effect sizes of differences between groups (Cohen's d) with 95% confidence intervals (95%CI) were 0.25 (95%CI 0.03–0.48) for C3bc, 0.33 (95%CI 0.09–0.57) for C5b-C9

**Data availability statement:** Data cannot be shared publicly according to Norwegian law regulations on research using human subjects where health information has been extracted. Relevant requests for access data need to approved by the Regional Committee for Medical Research Ethics North Norway and Patient Data Protection Office at Akerhus University Hospital in Norway. A written application for data access must be sent to Patient Data Protection Office at Akershus University Hospital. Email address: fellesmail.personvernombud@ahus.no.

**Funding:** The author(s) received no specific funding for this work.

**Competing interests:** The authors have declared that no competing interests exist.

terminal complex (TCC), 0.88 (95%CI 0.61–1.15) for C-reactive protein, and 1.64 (95%CI 1.37–1.91) for D-dimer. The areas under the curves derived from comparison receiver operating characteristics analyses with corresponding 95%CIs for C3bc, C5b-C9 terminal complex (TCC), C-reactive protein and D-dimer were 0.56 (95%CI 0.50–0.63), 0.64 (95%CI 0.58–0.69), 0.73 (95%CI 0.67–0.79) and 0.92 (95%CI 0.89–0.95), respectively.

## Conclusion

The plasma levels of the complement activation products C3bc and C5b-C9 terminal complex (TCC) were elevated in patients with acute deep vein thrombosis but displayed low diagnostic performance for deep vein thrombosis alone or together with D-dimer.

## Background

Deep vein thrombosis (DVT) is the formation of a blood clot in the deep veins of the body, most often the lower limbs, leading to unspecific symptoms such as redness, swelling and pain of the affected limb [1]. Early, precise diagnosis and treatment of patients with suspected DVT are important to restrict thrombus growth and adverse complications such as embolization and post-thrombotic syndrome, as well as to minimize added costs and unnecessary bleeding risk of antithrombotic treatment in patients without DVT [2,3].

Current diagnostic guidelines recommend using a multi-step diagnostic algorithm consisting of clinical pre-test probability assessment (PTP), the D-dimer blood test and radiological procedures, most often compression ultrasound (CUS), to ensure sufficient diagnostic precision and cost-effectiveness for patients with suspected DVT [3–5]. Patients are recommended to be categorized into low PTP or high PTP (simplified Wells score) during history taking and clinical examination, of which patients with low PTP should have their D-dimer levels measured. Patients with low PTP and D-dimer levels below cut-off value can safely be ruled out from further testing and treatment because of D-dimer's excellent negative predictive value of >95% for DVT in this patient group [3–5]. Although a single blood test is quick and easy to perform, D-dimer's low positive predictive value of 40% is not sufficient to confirm the DVT diagnosis. CUS has an excellent positive predictive value of over 95% in patients with low PTP and positive D-dimer and for patients with high PTP, and is therefore recommended as the final confirmatory test for DVT suspicion [3–5]. Although CUS does not entail contrast and radiation exposure such as CT venography, it is still time consuming and vulnerable for unavailable equipment and expertise which can lead to diagnostic delay [3,4]. Therefore, selecting patients in need of CUS using PTP assessments and D-dimer testing is pivotal to ensure the diagnostic algorithm's cost-effectiveness. When the algorithm for DVT is followed rigorously, it can reduce the number of required CUS procedures by up to 30–40% of all patients with suspected DVT [3,6,7].

Increasing evidence suggests low compliance to the abovementioned algorithm leading to unnecessary use of CUS due to bypassing of the PTP assessment and D-dimer based selection strategy. Excess use of CUS could lower the test's positive predictive value of DVT and increase health expenditures, threatening the algorithms diagnostic precision and cost-effectiveness [3–5]. In a recent systematic review, Kraaijpoel et al. reported that only 6–25% of the CUS performed confirmed a DVT [8], indicating low adherence to the diagnostic algorithm. Low adherence to the diagnostic algorithm is supported by data from the GARFIELD-VTE registry, including 8189 DVT-patients from 28 countries, where use of PTP and D-dimer in the diagnostic work-up was reported in only 4.6% and 25.7% of the patients, receptively [9]. Regardless of the underlying reasons for the high rates of performed CUS, it emphasizes potential room for improvement of the current diagnostic strategy for patients with suspected DVT.

Growing evidence support a comprehensive pathophysiological interplay between components of the coagulation and complement cascades [10]. Several clinical epidemiological studies have reported associations between elevated plasma levels of complement factors and their activation products and risk of venous thromboembolism (VTE) [11–14]. Relation to disease pathophysiology could be a key feature when aiming to discover novel diagnostic biomarkers for a disease [15]. As complement system activation is distinctly associated to acute DVT etiology through an acute inflammatory response [16] and complement activation occur upstream of inflammatory pathways [17], we hypothesized that complement activation products could potentially improve the diagnostic performance alone or together with D-dimer for the diagnosis of DVT. Therefore, we aimed to investigate the diagnostic utility of complement activation products, assessed by plasma C3bc and soluble C5b-9 terminal complement complex (TCC), and C-reactive protein (CRP), alone or together with D-dimer, for acute DVT in a cohort of outpatients referred to the emergency department with suspected DVT using CUS as reference standard.

## Methods

### Study design and setting

We performed a diagnostic observational cross-sectional study with prospective, consecutive patient sampling at Akershus University Hospital, Norway, from 18th of June 2021–26th of June 2023. The study setting was outpatients with suspected first-time DVT referred by primary health care physicians to either the emergency department or VTE outpatient clinic at Akershus University Hospital. This study was approved by the regional committee for health and research ethics (Project number: 200878) and complies with STARD (Standards for Reporting of Diagnostic Accuracy Studies) 2015 guidelines and the declaration of Helsinki. All potentially eligible patients received oral and written information about the study upon consenting to participate.

### Study population

Patients with signs and symptoms of lower extremity DVT, *i.e.*, swelling, redness, venous ectasia, and/or pain, using Akershus University Hospital as their local hospital who consented to study participation were considered potentially eligible patients. The patients were identified through referrals sent by primary health care or emergency department physicians for CUS to confirm or exclude DVT diagnosis. We applied no restrictions regarding pre-test probability or D-dimer results for study inclusion. Potentially eligible patients underwent a thorough eligibility assessment, where our pre-defined exclusion criteria were younger than 18 years or older than 80 years, life expectancy less than 2 years, inpatients, and where reference standard or index test sampling were not performed. Additionally, patients on ongoing anticoagulant therapy were excluded, whereas patients who received a single dose of anticoagulant drugs initiated by primary health care physicians before hospital arrival were included.

### Sampling of plasma index tests, D-dimer and CRP

The index tests of the study were plasma C3bc and soluble TCC. Non-fasting blood samples were collected in EDTA tubes (index tests, CRP) and citrate tubes (D-dimer) from an antecubital vein at the same time as routine blood work-up

was performed at the emergency department or at the VTE outpatient clinic within 24 hours of hospital arrival. The index tests and CRP samples were centrifuged in a swing-out centrifuge at 2500 g for 15 minutes. Plasma was thereafter transferred into labeled cryotubes and apportioned into 0.5mL aliquots, frozen immediately, and stored at −80 °C.

## Measurement of index tests, D-dimer and CRP

The samples were thawed on crushed ice and plasma levels of C3bc and soluble TCC were measured at Nordland Hospital in Bodø using an in-house enzyme-linked immunosorbent assay (ELISA) as described previously [18–20]. Results are given in complement arbitrary units (CAU)/mL. The inter assay coefficients of variance (CVs) of the in-house assays were 5.0% for C3bc and 5.2% for TCC, respectively. Plasma CRP was measured at Oslo University hospital by ELISA using matched antibodies (catalog #DY1707; R&D Systems, Stillwater, MN) in a 384 format using a combination of a CyBi SELMA (CyBio, Jena, Germany), EL406 washer/dispenser (Biotek, Winooski, VT), and Synergy H2 microplate reader (Biotek). The intra- and interassay coefficients of variation were <10%. Plasma D-dimer was analyzed immediately after sampling at the Department of Multidisciplinary Laboratory Medicine and Medical Biochemistry at Akershus University Hospital using the Tina-quant® D-dimer Gen. 2 assay on Cobas t711 (D-DI2; Roche Diagnostics).

## Reference standard

Whole-leg CUS examination of the deep veins of calf, thigh, and loin was accepted as reference standard in this study. We chose not to include CT venography as an additional reference standard to enable direct comparison of the index tests to the diagnostic performance of CUS. Criteria for CUS results were pre-defined and dichotomous: an uncompressible distal (*i.e.*, distal to vena poplitea) or proximal (*i.e.*, at the site of vena poplitea or more proximal veins) deep vein or intramuscular thrombus was considered a positive test result whereas compressible deep veins were a negative result. Furthermore, isolated superficial vein thrombosis (*i.e.,* uncompressible superficial veins and compressible deep veins) was a negative result. CUS was performed and interpreted by trained physicians before or after blood sample collection, within the first 24 hours after hospital arrival. CUS examinations and quantification of C3bc, TCC and CRP were executed at two different institutions at different timepoints and the personnel performing the procedures were blinded to each other's results.

## Disease of interest

Patients with confirmed first-time DVT of their lower limbs, proximal or distal, in need of at least 3 months of anticoagulation therapy were considered fulfilling the disease of interest criteria and were termed as DVT-patients. Patients with recurrent DVT or first-time DVT-patients in need of catheter-directed thrombolysis were excluded. DVT-patients with concurrent pulmonary embolism (PE) or superficial vein thrombosis were included, whereas patients with isolated PE were excluded. Patients with isolated superficial vein thrombosis were included as non-DVT patients (diagnostic comparison group). Patients with uncertain reference standard results in the emergency department or the VTE outpatient clinic were referred to another operator either at the emergency department or at the radiological department at Akershus University Hospital for a second and conclusive examination. To ensure the spectrum of differential diagnosis of DVT (*i.e.,* diagnoses with similar clinical presentation), no restrictions were applied regarding the final diagnoses of the non-DVT patients. We extracted the international classification of disease codes version 10 (ICD-10) of non-DVT patients to investigate this spectrum. Since the sensitivity of CUS may vary depending on patient-related factors and the examinators' experience, we investigated the proportion of potential CUS misclassification among non-DVT patient by searching in their hospital medical records for missed VTE (DVT and PE) diagnoses within three months after study inclusion.

## Data extraction

After retrieving consent, patients' medical records at Akershus University Hospital were reviewed for data extraction of radiological procedures, demographic- and clinical characteristics, disease presentation, comorbidities, medication usage,

results from standard admission blood work and D-dimer results. Definitions of extracted variables and diagnostic accuracy estimates are shown in S1 Table. Missing data from either index- or reference standard led to exclusion of patients from the study.

### Study outcomes

The main, predefined outcome of this study was area under the curve (AUC) with corresponding 95% confidence intervals (95%CI) for the index tests C3bc and TCC using CUS results as reference standard in receiver operating characteristics (ROC) analyses. We also performed ROC analyses for CRP and D-dimer to compare the complement products' performance to established markers of inflammation and coagulation, respectively. However, missed data of CRP and D-dimer did not lead to patient exclusion.

### Statistical analyses

To compare the plasma levels of C3bc, TCC, CRP and D-dimer between the two groups (acute DVT and non-DVT), we calculated the Cohen's d effect size, with corresponding 95% confidence interval estimated by bootstrapping of 5,000 permutations. Typically, a Cohen's d effect size of 0.2 would be interpreted as small, 0.5 as medium, and 0.8 as large [21]. Mann-Whitney U test was also applied to compare differences in plasma levels of the biomarkers between the two groups in addition to the continuous demographic baseline variables of the study population. The chi-square test was performed on categorical clinical and demographic variables. The significance level was a p-value $\leq 0.05$. Cohens d, nonparametric ROCs with its corresponding AUCs with 95%CI, p-values and graphical presentations were generated by Stata version 18.0 (Stata Corporation LP, College Station, TX, USA). Partial dependence plots using models with DVT predicted by D-dimer and either C3bc, TCC or CRP, respectively, were applied to investigate whether the combination of D-dimer with either C3bc, TCC or CRP could contribute to the predicted probability of DVT by D-dimer alone. The plots were generated using R (ver. 4.3.3, R Core Team, 2024) [22]. Sample size calculation was based on the ROC using C3bc and TCC as index tests on the entire patient cohort using the method described by Obuchowski [23], where we assumed AUCs of 0.7, DVT-prevalence of 30%, half-width of confidence interval to 0.07 and z-score at 1.96, which generated a need to include at least 278 patients referred to the emergency department with signs and symptoms of acute DVT.

## Results

### Flow of participants

The flow of excluded and included participants is illustrated in Fig 1. Four hundred and twelve potentially eligible patients were identified based on signs and symptoms of lower limb DVT with referrals to CUS, and of these, 35 were excluded by pre-defined exclusion criteria (Fig 1). Of the 377 eligible participants, 11 were excluded due to unforeseen reasons including failure to transport the samples to analyzing facility (n = 9), coagulation of index test sample (n = 1) and withdrawal of consent (n = 1). After all exclusions, a total of 366 patients were included in the study.

### Baseline demographics and clinical characteristics

The study sample of 366 participants consisted of 103 DVT-patients and 263 non-DVT-patients, yielding a DVT prevalence of 28.1%. The demographics and clinical characteristics of study participants are shown in Table 1. In the DVT-group, a larger proportion of the participants were male (63%) compared to the non-DVT-group (40%). The median age was 58 years in both groups and the participants had low rates (0–6%) of comorbidities such as myocardial infarction, previous cancer, chronic pulmonary disease, heart failure, stroke or connective tissue disorder. The complete list of

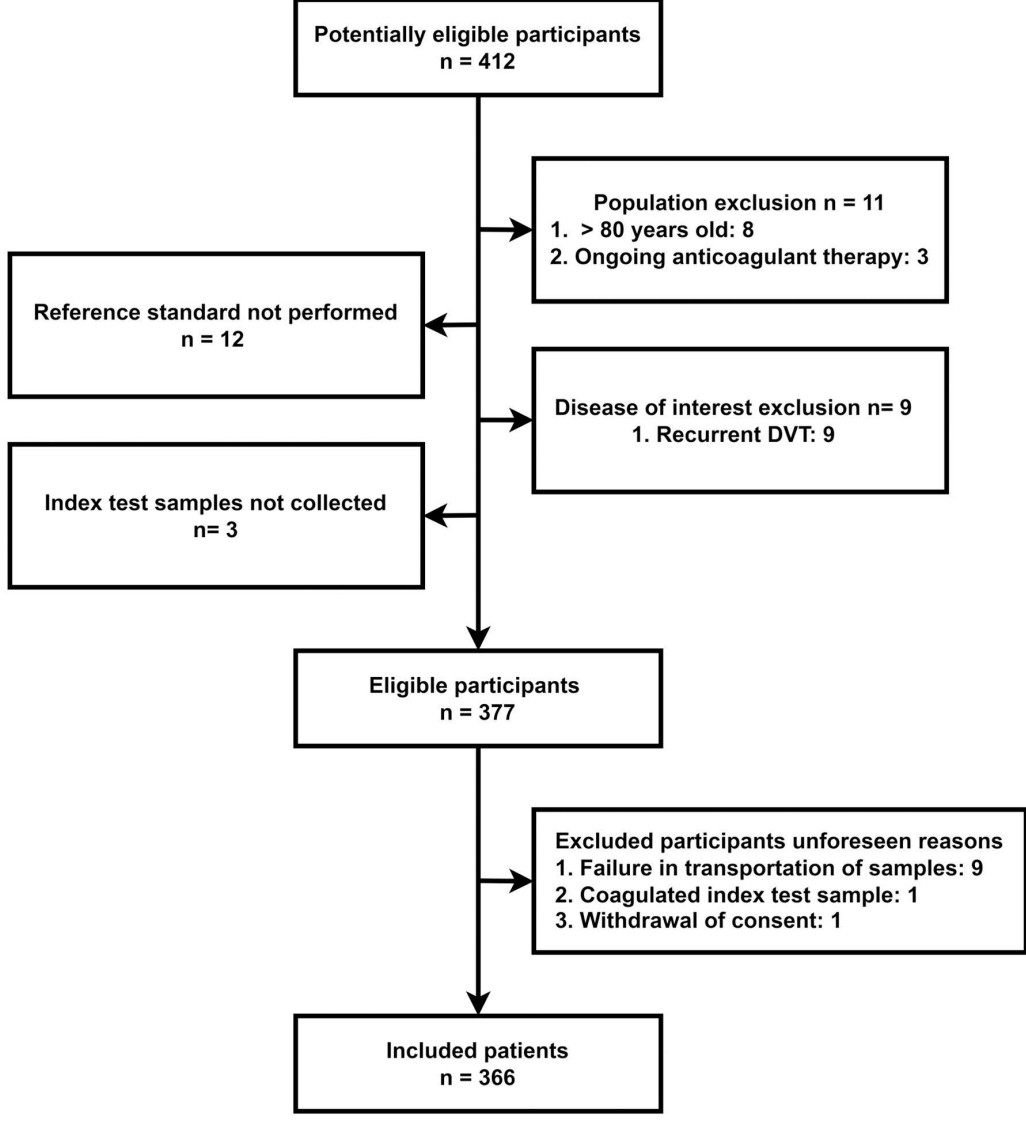

**Fig 1. Flow diagram of included and excluded patients.**

differential diagnoses of non-DVT-patients can be found in S2 Table. Most non-DVT patients received the diagnoses "pain in limb, calf/knee" or "rupture of popliteal cyst".

More than half of the participants reported symptoms of DVT for at least 98 hours before seeking medical attention from primary health care physicians. A single dose of anticoagulants was administered to 40% of DVT-patients and 23% of non-DVT patients before hospital arrival. Furthermore, a higher proportion of the DVT-patients had risk factors such as recent surgery/trauma, family history of VTE, and BMI ≥ 25 kg/m$^2$, while other risk factors such as active cancer, hormonal therapy, pregnancy related factors or known thrombophilia were similarly distributed in the two groups. Among the DVT-patients, 23% had concomitant PE and 72% had proximal DVTs. Among the non-DVT patients, three had evidence of missed DVT and one of missed PE, as these diagnoses were confirmed within 3 months after study inclusion date.

**Table 1. Demographic and clinical characteristics of included patients.**

| | DVT patients (n = 103) | Non-DVT patients (n = 263) | P-value |
|---|---|---|---|
| **Prevalence of DVT** | 28.1% | N/A | N/A |
| **Age (years), median (IQR)** | 58 (47-67) | 58 (49-70) | 0.31 |
| **Male, n (%)** | 65 (63%) | 106 (40%) | 0.09 |
| **Symptom duration > 98 h, n (%)** | 58 (56%) | 155 (59%) | 0.65 |
| **Symptom duration ≤ 48 h, n (%)** | 9 (9%) | 33 (13%) | 0.69 |
| **Surgery/trauma, n (%)** | 18 (17%) | 29 (11%) | 0.10 |
| **Family history of VTE (n, %)** | 18 (17%) | 20 (8%) | 0.02 |
| **Hormonal therapy, n (%)** | 6 (6%) | 12 (5%) | 0.62 |
| **Smoking, n (%)** | 9 (9%) | 30 (11%) | 0.43 |
| **BMI ≥ 25 kg/m$^2$, n (%)** | 36 (35%) | 68 (26%) | 0.13 |
| **Pregnancy related factors, n (%)** | 1 (1%) | 4 (2%) | 0.68 |
| **Known thrombophilia, n (%)** | 5 (5%) | 6 (2%) | 0.20 |
| **Cancer** | | | |
| Previous, n (%) | 6 (6%) | 16 (6%) | 0.93 |
| Active, n (%) | 6 (6%) | 23 (9%) | 0.35 |
| **Myocardial infarction, n (%)** | 5 (5%) | 13 (5%) | 0.92 |
| **COPD, n (%)** | 6 (6%) | 12 (5%) | 0.62 |
| **Heart failure, n (%)** | 1 (1%) | 3 (1%) | 0.89 |
| **Stroke, n (%)** | 2 (2%) | 5 (2%) | 0.98 |
| **Connective tissue disorder, n (%)** | 0 | 1 (0.4%) | 0.53 |
| **Antithrombotic therapy, n (%)** | | | |
| DOAC* | 5 (5%) | 10 (4%) | 0.65 |
| LMWH* | 36 (35%) | 51 (19%) | <0.00 |
| ADP receptor antagonist | 0 | 6 (2%) | 0.12 |
| Acetylsalicylic acid | 10 (10%) | 32 (12%) | 0.51 |
| **Isolated DVT, n (%)** | 79 (77%) | N/A | N/A |
| **DVT and PE, n (%)** | 24 (23%) | N/A | N/A |
| **Proximal DVT, n (%)** | 74 (72%) | N/A | N/A |
| **Missed VTEs** | | | |
| DVT, n (%) | N/A | 3 (1%) | N/A |
| PE, n (%) | N/A | 1 (0.4%) | N/A |
| DVT and PE, n (%) | N/A | 0 | N/A |

DVT: deep vein thrombosis, IQR: interquartile range, VTE: venous thromboembolism, BMI: body mass index, COPD: chronic obstructive pulmonary disease, DOAC: direct oral anticoagulants, LMWH: low molecular weight heparin, ADP: adenosine-diphosphate, PE; pulmonary embolism.

*Single dose of DOAC or LMWH awaiting diagnostic work-up.

## Plasma levels of C3bc, TCC, CRP and D-dimer in acute DVT

Plasma levels (medians with interquartile range) of C3bc, TCC, CRP, and D-dimer in DVT- and non-DVT patients are shown in Fig 2. In DVT and non-DVT patients the median plasma levels were 8.4 CAU/mL and 7.6 CAU/mL for C3bc, 0.44 CAU/mL and 0.36 CAU/mL for TCC, 44.8 mg/L and 15.1 mg/L for CRP, and 3.3 mg/L and 0.5 mg/L for D-dimer. The corresponding Cohen's d effect sizes were 0.25 (95% CI: 0.03–0.48) for C3bc, 0.33 (95% CI: 0.09–0.57) for TCC, 0.88 (95% CI: 0.61–1.15) for CRP, and 1.64 (95% CI: 1.37–1.91) for D-dimer (Table 2).

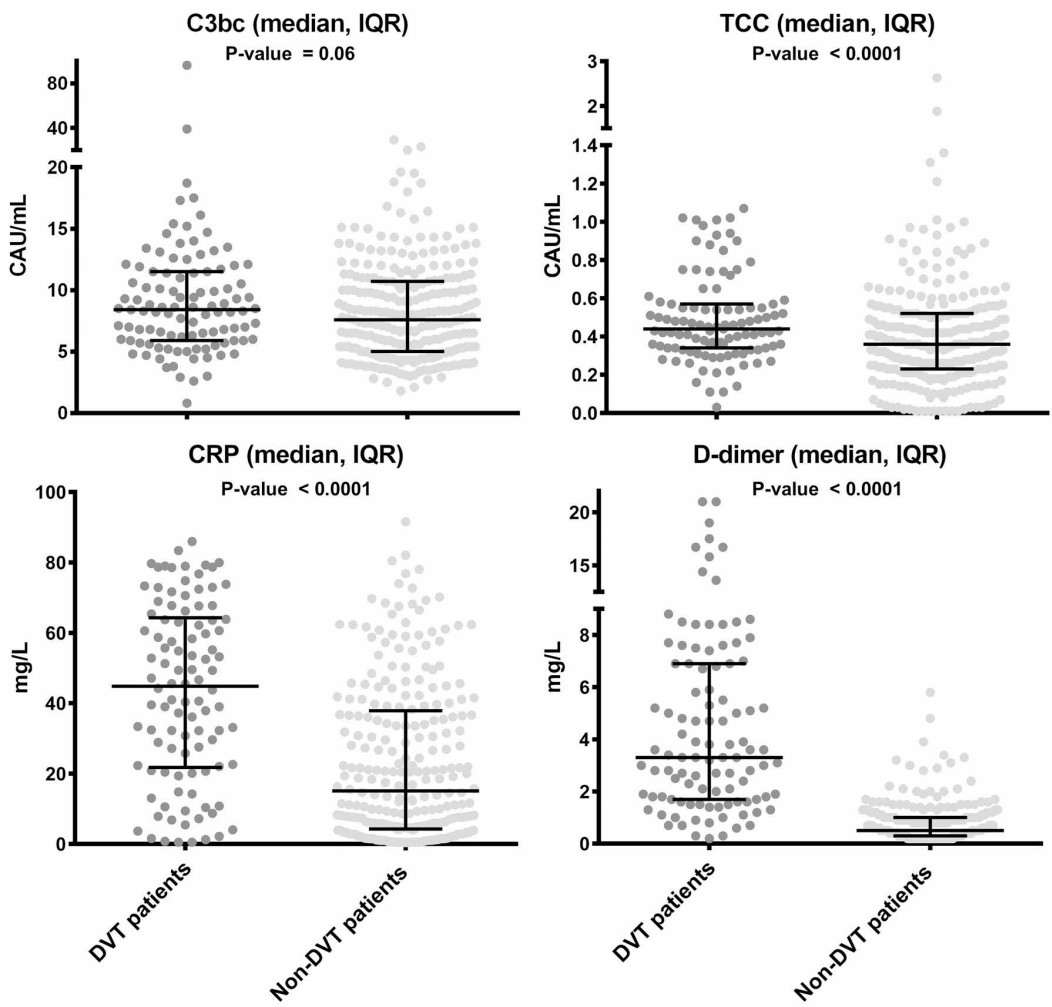

**Fig 2. Distribution of plasma concentration of A) C3bc, B) Terminal complement complex (TCC), C) C-reactive protein (CRP) and D) D-dimer with visualization of their corresponding median and interquartile range.**

## Diagnostic performance of C3bc, TCC, CRP and D-dimer

The ROC curves for C3bc, TCC, CRP and D-dimer are illustrated in Fig 3. The AUCs were 0.56 (95% CI: 0.50–0.63) for C3bc, 0.64 (95% CI: 0.58–0.69) for TCC, 0.73 (95% CI: 0.67–0.79) for CRP, and 0.92 (95% CI: 0.89–0.95) for D-dimer. The AUC of D-dimer was 0.92 (95% CI 0.89–0.95), and at the conventional cut-off value of ≥ 0.50 mg/L, D-dimer had a sensitivity of 97%, specificity of 42%, negative predictive value of 97% and positive predictive value of 40% (contingency table can be found in S3 Table). The partial dependence plots showed that predicted DVT was mainly dependent on D-dimer regardless of concentration variation of C3bc, TCC and CRP, respectively (Fig 4).

## Discussion

In this cross-sectional study with consecutive patient sampling, we investigated the diagnostic performance of the complement activation products C3bc and TCC, as well as CRP. Even though plasma levels of C3bc, TCC and CRP were higher in DVT-patients than non-DVT patients, the overall diagnostic performance of C3bc, TCC, and CRP were low to moderate

**Table 2. Cohen's d effect sizes with 95% confidence intervals (CI) for between group comparisons (DVT and non-DVT) of C3bc, TCC, CRP, and D-dimer.**

|  | Cohen's d | 95% CI |
|---|---|---|
| C3bc | 0.25 | 0.03–0.48 |
| TCC | 0.33 | 0.09–0.57 |
| CRP | 0.88 | 0.61–1.15 |
| D-dimer | 1.64 | 1.37–1.91 |

Abbreviations: CRP: C-reactive protein, DVT: Deep vein thrombosis, TCC: Terminal complement complex.

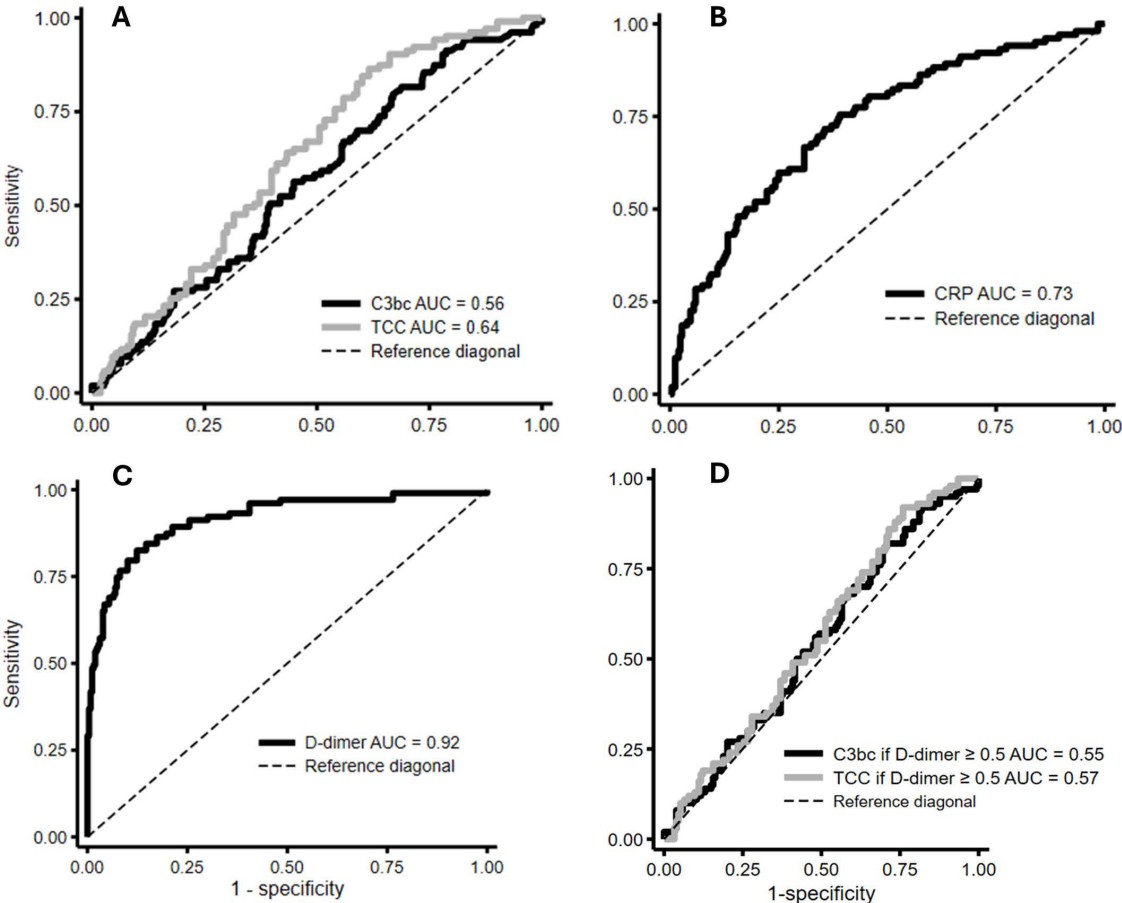

**Fig 3. Receiver operating characteristics (ROC) curves and their area under the curve (AUC) for A) C3bc and terminal complement complex (TCC), B) C-reactive protein (CRP), C) D-dimer and D) C3bc and TCC in patients with D-dimer ≥ 0.50 mg/L.**

with AUCs at 0.56, 0.64, and 0.73, respectively. The diagnostic performance of D-dimer in our cohort was as expected, with a high AUC (0.92) and sensitivity and negative predictive value of 97% for 0.50 mg/L as cut-off value. Partial dependence plots showed that predicted DVT was mainly dependent on D-dimer regardless of plasma levels of C3bc, TCC and CRP.

The diagnostic performances of the index tests C3bc and TCC were inferior to D-dimer and displayed limited ability to improve the current diagnostic "rule out selection strategy" for DVT. These results were also supported by our partial

## Partial dependence plots

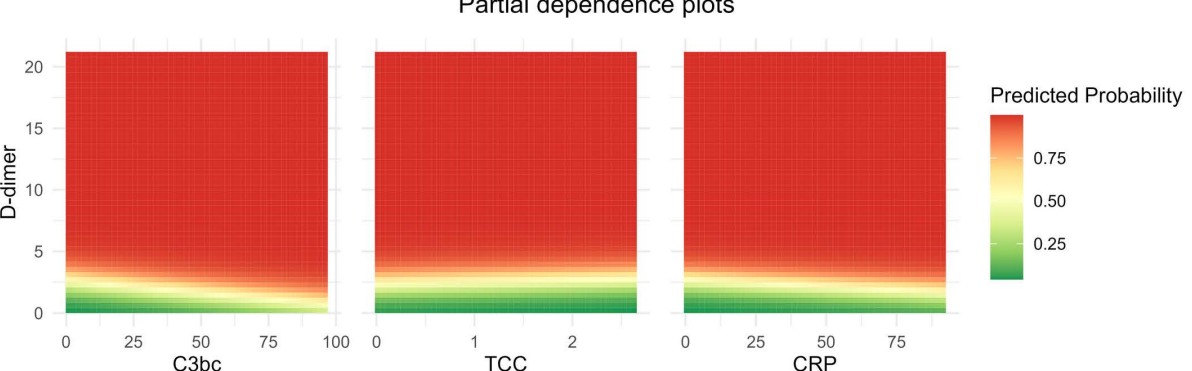

**Fig 4. Partial dependence plots using models were DVT is predicted by D-dimer and either C3bc, terminal complement complex (TCC) or C-reactive protein (CRP).** The predicted probability of deep vein thrombosis (DVT) appeared to mainly be dependent of D-dimer regardless of concentration variation of C3bc, TCC and CRP. A slight non-linear trend was observed for C3bc; however, this was most likely due to outlier values.

dependence plots, which showed that the predicted probability of DVT was mainly dependent on D-dimer and not C3bc or TCC. The low diagnostic performance of the index tests suggest that complement activation is not a specific process of DVT and occurs also in differential diagnoses of DVT, presumably because complement activation reflects initiation of a general inflammatory response. Accordingly, the index tests did not show potential as diagnostic tests for a "rule in selection strategy" for further testing or treatment of patients with suspected DVT.

Notably, the diagnostic performance of the markers, C3bc; TCC; CRP and D-dimer, improved down-stream in thrombo-inflammatory pathways. The complement system is part of innate immunity and is promptly activated up-stream of an inflammatory response through the classical-, lectin- and alternative pathways [24]. All complement pathways lead to formation of C3 convertase, which cleaves complement factor C3 into C3a and C3bc [24]. The latter activation product, C3bc, is a key mediator contributing to the formation of C5 convertase which eventually leads to the terminal complement complex (TCC), a complex end-product consisting of C5b-9 [24]. The down-stream inflammatory marker CRP becomes detectable in blood later in the immune response [25,26]. However, CRP can also contribute to complement activation by opsonization of pathogens activating and enabling binding of complement factor C1q which activates the classical pathway [27]. Accumulating evidence suggests a comprehensive crosstalk between components of the complement- and coagulation systems [28]. The improved diagnostic performance of activation products downstream in the thrombo-inflammatory process may indicate that the diagnostic performance of the inflammation markers is mediated through intertwining thrombo-inflammatory pathways where D-dimer serve as an end-product.

It has become increasingly evident during the last years that the complement system, and components of the lectin pathway, in particular, is implicated in the pathogenesis of VTE [11–13,29–31]. However, no study has investigated the diagnostic performance of complement factors or activation products for DVT. Recently, Iglesias et al. [32] reported moderately elevated plasma levels of complement factor H related 5 protein (CFHR5) in acute VTE compared to controls and replicated this finding in several case-control studies. CFHR5 facilitates activation of the alternative pathway of the complement system by inhibiting factor H, thereby contributing to the amplification loop with subsequent increased C3bc formation [33]. However, the case-control design using convenience sampling and matching is known to overestimate the diagnostic performance compared with a recommended cross-sectional design with clinically relevant comparison groups (*i.e.,* non-DVT patients with sign and symptoms suspicious of DVT) [34,35]. Thus, the clinical utility of CFHR5 as a diagnostic biomarker in patients with suspected DVT remains to be determined. Our findings indicate a distinction between the diagnostic power and involvement in the pathogenesis of VTE for components of the complement system and complement activation products.

The main strength of our study was the prospective consecutive sampling strategy including patients with suspected lower limb DVT to reduce the risk of selection bias. It is pivotal to avoid convenience sampling from two different source populations (*i.e.*, two-gate case-control sampling) to preserve the spectrum of disease severity of DVT as well as the spectrum of alternative/differential diagnoses to DVT [34]. If these spectrums are compromised, the risk of falsely increased sensitivity and specificity is high [34,36]. The distribution of patients with proximal- and distal DVT, DVT and PE, isolated DVTs and differential/alternative diagnoses of non-DVT patients were comparable to the reported distributions in clinical practice [37–39], indicating that disease spectrums were most likely preserved in our study. Still, the prevalence of DVT was somewhat higher than observed in clinical practice, indicating a chance of selection bias where the study cohort of non-DVT-patients are at risk of not being clinically representable. Such selection bias would most likely lead to a falsely high specificity and positive predictive value [40], thus, not impacting the overall negative results of our study. Comparison with only one reference standard (whole-leg CUS), and that the personnel assessing the index tests were unaware of the DVT status and vice versa, reduced the risk of information biases such as differential or partial verification bias [36]. Still, as whole-leg CUS can be difficult to interpret, especially for distal DVTs, the study is at risk of information bias, imperfect gold standard bias, which could falsely increase the index tests' sensitivity and negative predictive values [36]. However, only four non-DVT patients had a confirmed VTE within three months of inclusion, indicating a low risk of this bias on our main results. An alternative reference standard could have been CT venography, which has comparable diagnostic accuracy to CUS specificity and is less prone to patient-related factors (such as obesity) and examiner-related factors (such as less experience examiner) [41]. CT venography's are recommended in clinical guidelines only for a few very selected patients with DVT suspicion as it is more time-consuming than CUS and exposes patients to radiation and contrast medium-induced morbidity [3]. As no other study has investigated the diagnostic utility of complement activation products in the setting of DVT suspicion, we aimed to discover the discriminatory abilities of C3bc and TCC in an unselected group of patients with suspected DVT. Hence, we refrained from investigating the performance of the biomarkers within framework of the recommended diagnostic algorithm of DVT. This is a core limitation of the study; the diagnostic accuracy of the investigated biomarkers cannot be interpreted directly into context of current diagnostic guidelines of DVT suspicion. Furthermore, the generalizability of the study results is limited. First, including only first-time lower-limb DVT reduces the transferability of the results to patients with recurrent DVT and patients with DVT at atypical sites. Second, this study is a single-center study which limits the sample size and exclude differences between clinical practices in various hospitals and countries.

In conclusion, even though markers of complement activation (C3bc and TCC) were slightly elevated in acute DVT, they lacked discriminatory power individually and in combination with D-dimer to be useful in the diagnostic work-up of patients with suspected lower-limb DVT.

## Supporting information

**S1 Table. Definitions of extracted variables and diagnostic accuracy estimates.**
(DOCX)

**S2 Table. Registered diagnoses of non-deep vein thrombosis patients.** List and frequency of all international classification of diseases version 10 (ICD-10) registered in Akershus university hospital medical record system of patients in whom deep vein thrombosis were excluded.
(DOCX)

**S3 Table. Contingency table of D-dimer.** Contingency table displaying diagnostic accuracy estimates of D-dimer using the cut-off value of ≥ 0.50 mg/L.
(DOCX)

## Author contributions

**Conceptualization:** John-Bjarne Hansen, Ellen Elisabeth Brodin.

**Data curation:** Øyvind Øverli, Tom Eirik Mollnes, Annika Elisabet Michelsen, Gholamreza Jafari Yeganeh.

**Formal analysis:** Vårin Eiriksdatter Wikan, Thor Ueland, Tom Eirik Mollnes, Annika Elisabet Michelsen.

**Investigation:** Vårin Eiriksdatter Wikan, Øyvind Øverli, Sigrid Kufaas Brækkan, Gholamreza Jafari Yeganeh, John-Bjarne Hansen.

**Methodology:** Vårin Eiriksdatter Wikan, John-Bjarne Hansen, Ellen Elisabeth Brodin.

**Project administration:** Ellen Elisabeth Brodin.

**Supervision:** Sigrid Kufaas Brækkan, John-Bjarne Hansen.

**Visualization:** Vårin Eiriksdatter Wikan, Sigrid Kufaas Brækkan.

**Writing – original draft:** Vårin Eiriksdatter Wikan, Øyvind Øverli, Thor Ueland, Tom Eirik Mollnes.

**Writing – review & editing:** Sigrid Kufaas Brækkan, Gholamreza Jafari Yeganeh, John-Bjarne Hansen, Ellen Elisabeth Brodin.

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
