## [Decision Letter · Decision Letter 0]

19 Apr 2025

PONE-D-25-03459Complement activation assessed by C3bc and C5b-9 terminal complex as diagnostic biomarkers for deep vein thrombosisPLOS ONE

Dear Dr. Wikan,

Thank you for submitting your manuscript to PLOS ONE. After careful consideration, we feel that it has merit but does not fully meet PLOS ONE’s publication criteria as it currently stands. Therefore, we invite you to submit a revised version of the manuscript that addresses the points raised during the review process.

We look forward to receiving your revised manuscript.

Kind regards,

Eyüp Serhat Çalık

Academic Editor

PLOS ONE

Journal Requirements:

2. In the online submission form, you indicated that data cannot be shared publicly according to Norwegian law regulations on research using human subjects where health information has been extracted. Relevant requests for access data need to approved by the Regional Committee for Medical Research Ethics North Norway and Patient Data Protection Office at Akerhus University Hospital in Norway.

4. Please remove all personal information, ensure that the data shared are in accordance with participant consent, and re-upload a fully anonymized data set.

5. We are unable to open your Supporting Information file [Letter of confirmation REK]. Please kindly revise as necessary and re-upload.

Additional Editor Comments :

I would like to congratulate the authors for this interesting study. I think it is important to investigate the diagnostic value of plasma levels of complement activation products C3bc and C5b-C9 terminal complex (TCC) in acute deep vein thrombosis. The manuscript was reviewed by two reviewers and their recommendations are as follows. We look forward to your point-by-point responses to the suggestions and to your re-submission of the manuscript with appropriate revisions. Good luck.

Reviewers' comments:

Reviewer's Responses to Questions

**Comments to the Author**

1. Is the manuscript technically sound, and do the data support the conclusions?

Reviewer #1: Yes

Reviewer #2: Yes

2. Has the statistical analysis been performed appropriately and rigorously? 

Reviewer #1: Yes

Reviewer #2: Yes

3. Have the authors made all data underlying the findings in their manuscript fully available?

Reviewer #1: Yes

Reviewer #2: No

4. Is the manuscript presented in an intelligible fashion and written in standard English?

Reviewer #1: Yes

Reviewer #2: Yes

5. Review Comments to the Author

Reviewer #1: Review Comments on the Manuscript: "Complement activation assessed by C3bc and C5b-9 terminal complex as diagnostic biomarkers for deep vein thrombosis"

This manuscript investigates the potential of C3bc and C5b-9 terminal complex (TCC) as diagnostic biomarkers for deep vein thrombosis (DVT) by examining 366 patients. The study demonstrates that the levels of complement activation products C3bc and C5b-9 TCC are elevated in acute DVT. However, their diagnostic performance for DVT, either alone or in combination with D-dimer, is found to be relatively low. The study design is rigorous, but several issues need to be addressed to improve the manuscript:

Confusing Background Section: The background section provides a rather disorganized description of the current diagnostic methods for DVT. It is recommended to optimize and summarize the existing diagnostic approaches and their limitations more clearly. This would help readers better understand the significance of the current study.

Missing P-values in Table 1: In Table 1, which presents the demographic and clinical characteristics of included patients, there is a lack of P-values comparing the DVT group and the Non-DVT group. It is essential to add a column for P-values to indicate whether there are statistically significant differences between the two groups for each parameter.

P-value Annotation in Figure 2: For the results of plasma levels of C3bc, TCC, CRP, and D-dimer in acute DVT presented in Figure 2, it is important to annotate the P-values. For example, use "ns" for non-significant differences, "*" for p<0.05, or directly label the exact P-values.

Limitation of Single-Center Study: The study is conducted in a single center, which inherently limits its generalizability. This limitation should be discussed in the manuscript.

Additional Limitations in the Discussion Section: The discussion section should include other limitations of the study. The authors should critically evaluate potential biases, sample size issues, or other factors that might have influenced the results.

Further Discussion on Complement Activation Products: Based on the findings of this study and relevant literature, the authors should provide a more in-depth discussion on why the plasma levels of C3bc and C5b-C9 TCC were elevated in acute DVT but exhibited poor diagnostic performance for DVT, either alone or in combination with D-dimer. This section should explore potential biological reasons and the implications for future research.

Overall, the study provides valuable insights into the role of complement activation in DVT. However, addressing the above concerns will enhance the clarity, rigor, and impact of the manuscript.

Reviewer #2: Dear Authors,

The work is an interesting attempt to define new markers of deep vein thrombosis. The relationship between complement components and the coagulation system is becoming more and more precisely known. Although the authors showed that plasma levels of the complement activation products C3bc and C5b-C9 terminal complex (TCC) were elevated in acute deep vein thrombosis, but displayed low diagnostic performance for deep vein thrombosis alone or with D-dimer.

The work is methodically correctly designed, and well written, however, I have a few doubts:

1. why patients' life-expectancy less than 2 years was one of exclusion criteria - why was this time chosen?

2. it is not cleared if missed VTE was searched for in all patients after inclusion in the study

kind regards

6. PLOS authors have the option to publish the peer review history of their article (what does this mean? ). If published, this will include your full peer review and any attached files.

**Do you want your identity to be public for this peer review?** For information about this choice, including consent withdrawal, please see our Privacy Policy .

Reviewer #1: No

Reviewer #2: No

---

## [Author Response · Author response to Decision Letter 1]

18 Jul 2025

We have uploaded a word-file the the responses as well.

Responses to comments from the editor:

Response: We have edited manuscript, supporting information and figure-file names according to PLOS ONE’s style requirements.

2. In the online submission form, you indicated that data cannot be shared publicly according to Norwegian law regulations on research using human subjects where health information has been extracted. Relevant requests for access data need to approved by the Regional Committee for Medical Research Ethics North Norway and Patient Data Protection Office at Akerhus University Hospital in Norway.

4. Please remove all personal information, ensure that the data shared are in accordance with participant consent, and re-upload a fully anonymized data set.

Response to comment 2, 3 and 4: Thank you for this comment. We acknowledge PLOS ONE’s effort to make medical research more transparent through requesting all underlying data to be made available when it is ethically and legally possible. Unfortunately, as described upon submission, we cannot make out data publicly available for legal reasons. Norwegian laws regarding patients’ privacy in medical research are rigorous. During the resubmission process we contacted the Patient Data Protection Office at Akershus University Hospital to investigate if an anonymized part of the underlying data could be published with the manuscript, but this request was rejected. However, the data protection office will allow written applications for access to the underlying data if the reader want to reproduce our research findings. We have therefore added a section in the manuscripts informing the reader about this possibility and to whom they can send the data request:

Added to the manuscript at page 20 line 400 to 403:

“Availability of underlying data

Underlying data of this study can be made available for readers who seek to reproduce this study’s results. A written application for data access must be sent to principal investigator Ellen Elisabeth Brodin. Email address: Ellen.Elisabeth.Brodin@ahus

We hope this information is sufficient for PLOS ONE to make an exemption from our policies regarding data accessibility.

5. We are unable to open your Supporting Information file [Letter of confirmation REK]. Please kindly revise as necessary and re-upload.

Response: We are sorry for the inconvenience. The re-uploaded file of “Letter of confirmation REK” is now in a standard pdf-format and should be readable.

Response: The captions for Supporting Information can now be found at the end of the manuscript after references in accordance with PLOS ONE’s guidelines:

Added to the manuscript at page 23-24 line 533 to 540:

“ Supporting information

S1 Table. Definitions of extracted variables and diagnostic accuracy estimates.

S2 Table. Registered diagnoses of non-deep vein thrombosis patients. List and frequency of all international classification of diseases version 10 (ICD-10) registered in Akershus university hospital medical record system of patients in whom deep vein thrombosis was excluded.

S3 Table. Contingency table of D-dimer. Contingency table displaying diagnostic accuracy estimates of D-dimer using the cut-off value of ≥ 0.50 mg/L.”

Response: Thank you for this reminder; we have checked the appearance of the references. No references have been added or removed.

Responses to comments from reviewer 1:

Reviewer #1: Review Comments on the Manuscript: "Complement activation assessed by C3bc and C5b-9 terminal complex as diagnostic biomarkers for deep vein thrombosis"

This manuscript investigates the potential of C3bc and C5b-9 terminal complex (TCC) as diagnostic biomarkers for deep vein thrombosis (DVT) by examining 366 patients. The study demonstrates that the levels of complement activation products C3bc and C5b-9 TCC are elevated in acute DVT. However, their diagnostic performance for DVT, either alone or in combination with D-dimer, is found to be relatively low. The study design is rigorous, but several issues need to be addressed to improve the manuscript:

1. Confusing Background Section: The background section provides a rather disorganized description of the current diagnostic methods for DVT. It is recommended to optimize and summarize the existing diagnostic approaches and their limitations more clearly. This would help readers better understand the significance of the current study.

Response: Thank you for this recommendation. We have rewritten the second and third section of the manuscript to give a more organized and comprehensive description of the current diagnostic algorithm of DVT and its limitations.

Rewritten parts of the introduction of manuscript at page 4 and 5 line 60 to 90:

“Current diagnostic guidelines recommend using a multi-step diagnostic algorithm consisting of clinical pre-test probability assessment (PTP), the D-dimer blood test and radiological procedures, most often compression ultrasound (CUS), to ensure sufficient diagnostic precision and cost-effectiveness for patients with suspected DVT [3-5]. Patients are recommended to be categorized into low PTP or high PTP (simplified Wells score) during history taking and clinical examination, of which patients with low PTP should have their D-dimer levels measured. Patients with low PTP and D-dimer levels below cut-off value can safely be ruled out from further testing and treatment because of D-dimer’s excellent negative predictive value of >95% for DVT in this patient group [3-5]. Although a single blood test is quick and easy to perform, D-dimer's low positive predictive value of under 50% is not sufficient to confirm the DVT diagnosis. CUS has an excellent positive predictive value of over 95% in patients with low PTP and positive D-dimer and for patients with high PTP, and is therefore recommended as the final confirmatory test for DVT suspicion [3-5]. Although CUS does not entail contrast and radiation exposure such as CT venography, it is still time consuming and vulnerable for unavailable equipment and expertise which can lead to diagnostic delay [3, 4]. Therefore, selecting patients in need of CUS using PTP assessments and D-dimer testing is pivotal to ensure the diagnostic algorithm’s cost-effectiveness. When the algorithm for DVT is followed rigorously, it can reduce the number of required CUS procedures by up to 30-40% of all patients with suspected DVT [3, 6, 7].

Increasing evidence suggests low compliance to the abovementioned algorithm leading to unnecessary use of CUS due to bypassing of the PTP assessment and D-dimer based selection strategy. Excess use of CUS could lower the test’s positive predictive value of DVT and increase health expenditures, threatening the algorithms diagnostic precision and cost-effectiveness [3-5]. In a recent systematic review, Kraaijpoel et al. reported that only 6-25% of the CUS performed confirmed a DVT [8], indicating low adherence to the diagnostic algorithm. Low adherence to the diagnostic algorithm is supported by data from the GARFIELD-VTE registry, including 8189 DVT-patients from 28 countries, where use of PTP and D-dimer in the diagnostic work-up was reported in only 4.6% and 25.7% of the patients, receptively [9]. Regardless of the underlying reasons for the high rates of performed CUS, it emphasizes potential room for improvement of the current diagnostic strategy for patients with suspected DVT.”

2. Missing P-values in Table 1: In Table 1, which presents the demographic and clinical characteristics of included patients, there is a lack of P-values comparing the DVT group and the Non-DVT group. It is essential to add a column for P-values to indicate whether there are statistically significant differences between the two groups for each parameter.

3. P-value Annotation in Figure 2: For the results of plasma levels of C3bc, TCC, CRP, and D-dimer in acute DVT presented in Figure 2, it is important to annotate the P-values. For example, use "ns" for non-significant differences, "*" for p<0.05, or directly label the exact P-values.

Response to comment 2 and 3: In accordance with your suggestions, we have added p-values to table 1 (page 12 and 13) and added p-values to TCC, CRP and D-dimer in Figure 2. In addition, we have added which type of significance test we have added and level of significance in the statistical analyses section.

We added this text to the last section of page 10 line 209 to 214:

“Whitney U-test was also applied to compare differences in plasma levels of the biomarkers between the two groups in addition to the continuous demographic baseline variables of the study population. The chi-square test was performed on categorical clinical and demographic variables. The significance level was a p-value ≤ 0.05.”

4. Limitation of Single-Center Study: The study is conducted in a single center, which inherently limits its generalizability. This limitation should be discussed in the manuscript.

5) Additional Limitations in the Discussion Section: The discussion section should include other limitations of the study. The authors should critically evaluate potential biases, sample size issues, or other factors that might have influenced the results.

Response to comment 4 and 5: Thank you for these remarks, we acknowledge the importance of communicating the limitations of the study comprehensively and clearly. Our intent with the second last section of the discussion was to point out both strengths and weaknesses. However, reading through it again, it is apparent that the description of the study’s weaknesses has been too vaguely described.

We have therefore rewritten parts of the discussion section to highlight weaknesses of our study design. Page 18 to 19 line 354 to 381:

“The main strength of our study was the prospective consecutive sampling strategy including patients with suspected lower limb DVT to reduce the risk of selection bias. It is pivotal to avoid convenience sampling from two different source populations (i.e., two-gate case-control sampling) to preserve the spectrum of disease severity of DVT as well as the spectrum of alternative diagnoses to DVT [34]. If these spectrums are compromised, the risk of falsely increased sensitivity and specificity is high [34, 36]. The distribution of patients with proximal- and distal DVT, DVT and PE, isolated DVTs and differential/alternative diagnoses of non-DVT patients were comparable to the reported distributions in routine clinical practice [37-39], indicating that disease spectrums were most likely preserved in our study. Still, the prevalence of DVT was somewhat higher than observed in clinical practice, indicating a chance of selection bias where the study cohort of non-DVT-patients are at risk of not being clinically representable. Such selection bias would most likely lead to a falsely high specificity and positive predictive value [40], thus, not impacting the overall negative results of our study. Comparison with only one reference standard (whole leg CUS), and that the personnel assessing the index tests were unaware of the DVT status and vice versa, reduced the risk of information biases such as differential or partial verification bias [36]. Still, as whole leg CUS can be difficult to interpret, especially for distal DVTs, the study is at risk of information bias, imperfect gold standard bias, which could falsely increase the index tests’ sensitivity and negative predictive values [36]. However, only four non-DVT patients had a confirmed VTE within three months of inclusion, indicating a low risk of this bias on our main results. An alternative reference standard could have been CT venography, which has comparable diagnostic accuracy to CUS specificity and is less prone to patient-related factors (such as obesity) and examiner-related factors (such as less experience examiner) [41]. CT venography’s are recommended in clinical guidelines only for a few very selected patients with DVT suspicion as it is more time-consuming than CUS and exposes patients to radiation and contrast medium-induced morbidity [3]. As no other study has investigated the diagnostic utility of complement activation products in the setting of DVT suspicion, we aimed

---

## [Editor Report · Decision Letter 1]

10 Sep 2025

Complement activation assessed by C3bc and C5b-9 terminal complex as diagnostic biomarkers for deep vein thrombosis

PONE-D-25-03459R1

Dear Dr. Wikan,

We’re pleased to inform you that your manuscript has been judged scientifically suitable for publication and will be formally accepted for publication once it meets all outstanding technical requirements.

Kind regards,

Eyüp Serhat Çalık

Academic Editor

PLOS ONE
---

## [Editor Report · Acceptance letter]

PONE-D-25-03459R1

PLOS ONE

Dear Dr. Wikan,

I'm pleased to inform you that your manuscript has been deemed suitable for publication in PLOS ONE. Congratulations! Your manuscript is now being handed over to our production team.

Kind regards,

on behalf of

Dr. Eyüp Serhat Çalık

Academic Editor

PLOS ONE